# Effects of Feeding Newly Hatched Larvae on the Growth, Survival, and Growth Patterns of Kawakawa (*Euthynnus affinis*) Larvae and Juveniles

**DOI:** 10.3390/ani15131997

**Published:** 2025-07-07

**Authors:** Lynn Nuruki, Aki Miyashima, Yasuo Agawa, Yoshifumi Sawada

**Affiliations:** 1Graduate School of Agriculture, Kindai University, Nakamachi 3327-204, Nara 631-8505, Nara, Japan; 2433660019t@nara.kindai.ac.jp; 2Aquaculture Research Institute, Kindai University, Oshima 1790-4, Kushimoto 649-3633, Wakayama, Japan; aki.miyashima@itp.kindai.ac.jp (A.M.); agawa@kindai.ac.jp (Y.A.)

**Keywords:** *Euthynnus affinis*, larvae, juveniles, fingerling production, feeding of newly hatched larvae, relative growth

## Abstract

The results of this study demonstrate that feeding live hatched larvae is essential for rearing larval and juvenile kawakawa (*Euthynnus affinis*), where there are no alternative feeds hatched larvae at present. Using hatched larvae of other species, such as *Oplegnathus fasciatus*, as feed for Kawakawa larvae and juveniles is a form of biomimicry, simulating their natural feeding habits. However, supplying large quantities of hatched larvae is labor-intensive and costly. Therefore, it is necessary to develop a formulated diet that can serve as an alternative to hatched larvae, which will require detailed nutritional studies. Furthermore, the analysis of relative growth in Kawakawa larvae and juveniles obtained in this study, along with comparisons to other tuna species, is expected to contribute significantly to advancing full-cycle aquaculture techniques for tuna. It also offers valuable insights into the ecological understanding of these species.

## 1. Introduction

Kawakawa, a small scombrid species, is broadly distributed across warm marine environments, notably the Indian Ocean and the western Pacific Ocean, including surrounding oceanic islands [1,2,3]. It is commercially exploited and consumed as a food resource [4], and small pelagic species—particularly small tunas such as *E. affinis* and its closely related congener, little tunny, *E. alletteratus*—constitute a significant proportion of global capture fisheries products [5].

In Japan, kawakawa has attracted attention due to its high fat content and high market potential as the material of Sushi and Sashimi [6]. To meet its demand, interest in kawakawa aquaculture production using wild-caught [7] and hatchery-reared juveniles has grown since 2012 in Japan [8]. Full-cycle aquaculture systems have been developed at fisheries research institutions, including those in Ehime Prefecture [9] and Wakayama Prefecture [10] in Japan. As an advantage of full-cycle aquaculture, it can be reared from fertilized eggs to maturity within a year, that facilitates selective breeding and husbandry protocols similar to those for other tunas that have already been commercialized [11]. At present, kawakawa farming has been established at an industrial scale, although current production remains limited to several tons, and in order to increase the aquaculture production of kawakawa, it is necessary to further improve its full-cycle aquaculture technology.

Scombrid larval and juvenile rearing techniques have been reported for Pacific bluefin tuna (*Thunnus orientalis*; [12,13,14,15]), Atlantic bluefin tuna (*Thunnus thynnus*; [16]), southern bluefin tuna (*Thunnus maccoyii*, [17]), yellowfin tuna (*Thunnus albacares*; [18,19,20,21]), striped bonito (*Sarda orientalis*; [22]), Atlantic bonito (*Sarda sarda*; [23]), kawakawa [24,25], Spanish mackerel (*Scomberomorus niphonius*; [26]), frigate tuna (*Auxis thazard*; [27]), bullet mackerel (*Auxis rocheri rocheri*; [28]), chub mackerel (*Scomber japonicas*; [29]), and Atlantic mackerel (*Scomber scombrus*; [30]). Among these species, complete aquaculture, encompassing the entire life cycle under controlled conditions, has been achieved for Pacific bluefin tuna [31], Atlantic bluefin tuna [32], and kawakawa [33,34]. These studies have been elucidated the common characteristics and challenges in the early development of scombrid species, including rapid growth [4,11,35] and an early onset of piscivory [10,11]. This early piscivory often leads to severe cannibalism, posing challenges for larval rearing. In kawakawa, strong piscivorous behavior, which has been reported on around 12 dph [11], making it a critical challenge in hatchery production. To mitigate cannibalism, strategies such as reducing stocking density [36], as larger individuals tend to prey on smaller ones [37], have been proposed. For Pacific bluefin tuna, minimizing size variation within reared juveniles has been recommended [38,39]. Besides this, survival rates have been improved by using slit-based size grading for kawakawa [33]. Additionally, feeding newly hatched larvae of other fish species has been considered effective in mitigating cannibalism [31,39]. However, for kawakawa, the optimal timing and quantity in hatched larval feeding remain unclear, although newly hatched larvae are also used in the rearing methods previously mentioned in the institutes in Ehime [9] and Wakayama Prefectures [10].

In fish larvae and juveniles, morphological change in each body part is remarkable as they grow. For example, structures essential for locomotion and feeding, such as the caudal fin, eyes, and mouthparts, tend to develop during the early stages. In contrast, musculature involved in swimming, as well as fins responsible for postural control—namely the dorsal, ventral, and pelvic fins—develop at later stages. Understanding such relative growth patterns—defined as changes in the growth rates of specific body parts in relation to overall body length, including transitions among positive allometry, isometry, and negative allometry—is crucial for optimizing rearing techniques by providing insights into functional transformations across developmental stages. Among such body parts, the mouth size and the jaw development are directly connected to the size of food that can be fed, which determines the suitable size and timing of the feeds fed to them. As other important matters, growth inflection points in relative growth during larval and juvenile stages are known to correspond to physiological and behavioral changes [15,40,41,42,43], which serves as a reference for developing their rearing protocols.

Relative growth patterns in scombrid larvae and juveniles have been well-documented for Pacific bluefin tuna [15], yellowfin tuna [20], striped bonito [25], and chub mackerel [44], contributing to improvements in larval rearing techniques. However, while reports on larval and juvenile rearing techniques for kawakawa exist, detailed studies on their absolute and relative growth patterns remain scarce.

This study aimed to clarify the necessity of feeding newly hatched larvae of other fish species to kawakawa larvae and important points for its improvement. To achieve this, experimental rearing trials were conducted using different feeding regimes: one group was fed rotifers (*Brachionus* spp.) and a formulated diet, while another group received additional hatched larvae feed from other fish species. The effects of these feeding strategies on the survival and growth of kawakawa larvae were evaluated. Additionally, specimens collected from commercial-scale seed production tanks were analyzed to assess their development, absolute growth, and relative growth patterns.

## 2. Materials and Methods

### 2.1. Ethical Approval

All experimental procedures involving animals were conducted in accordance with the Animal Experiment Regulations of Kindai University. The experimental protocol was approved by the Animal Experiment Committee of the Aquaculture Research Institute, Kindai University (Approval No. ARIKU-AEC-2022-23).

### 2.2. Experiment 1: Effects of Feeding Newly Hatched Larvae on the Growth and Survival of Kawakawa Larvae

#### 2.2.1. Test Fish and Rearing Methods

The test fish used in this experiment were obtained from fertilized eggs produced by hormone-induced spawning of kawakawa broodstock using HCG (500 IU/kgBW, GONATROPIN^®^3000, ASKA Animal Health Co., Ltd., Tokyo, Japan). This rearing of kawakawa larvae was carried out at Kindai University as part of a commercially oriented project. The normal hatching rate of the fertilized eggs was 83.2%.

Larvae were reared at 26 °C, and during the rearing period, water quality was measured daily at 08:00 and 13:00. The dissolved oxygen (DO) levels measured using a DO meter (550A, YSI, Yellow Springs, OH, USA), ranged from 83.6% to 107.6%. The pH values (D-24, HORIBA, Kyoto, Japan) were between 8.1 and 8.4, while salinity levels (EC300, YSI, Cheverly, MD, USA) ranged from 31.3 to 32.4 ppt, all within appropriate ranges. Furthermore, throughout the experiment, the smart larvae exhibited no signs of disease.

The rearing water temperature was maintained at 26.0 °C throughout the experiment. An illuminance of 2000 lx was provided by fluorescent, with a photoperiod of 13L:11D (light from 07:00 to 20:00). The aeration rate was set at 150 mL/min each during the daytime and two air stones (350 mL/min each) were added at night across all experimental tanks.

Kawakawa larvae were fed rotifers from 3 dph until the end of the experiment. The initial feeding density was 10 individuals/mL at 3 dph, where rotifers were supplied at 08:00 a.m. every morning and added to maintain the target density, and the feeding amount was increased in accordance with larval growth. The rotifers were enriched with EPA- and DHA-fortified freshwater Chlorella (Hyper Gloss, Marinetech Co., Ltd., Aichi, Japan; Super Fresh Chlorella V12, Chlorella Industry Co., Tokyo, Japan), taurine (Taurine CJ, Japan Nutrition Co., Tokyo, Japan), and DHA-algae-concentrated enrichment agents for rotifers (Marin Gross EX, Marine Tech Co., Aichi, Japan). From 9 days post-hatching (dph), in addition to rotifers, each tank was supplied with 10,000 newly hatched larvae of striped beakfish once every morning, obtained from the Kindai University Aquaculture Technology and Production Center Oshima Branch, as well as a formulated diet fed at any time (Magokoro, Nisshin Marubeni Feed Co., Tokyo, Japan). as well as a formulated diet (Magokoro, Nisshin Marubeni Feed Co., Tokyo, Japan). The seawater used for rearing was UV-sterilized and filtered, with the addition of with EPA- and DHA-concentrated Chlorella (Super Fresh Chlorella spV-12, Chlorella Industry Co., Tokyo, Japan) to maintain a density of 50 × 10^4^ cells/mL, adjusted according to the water supply volume.

Additionally, a water quality conditioner (Fish Green, Green Culture Co., Toyama, Japan) was applied at a rate of 15 g per tank before stocking eggs, and another 15 g per tank was added daily after feeding. Surface skimming to remove oil films was conducted at 08:30, 12:00, 15:00, and 16:30 across all experimental tanks.

#### 2.2.2. Experimental Groups

From 1 dph to 8 dph, all tanks were maintained under identical conditions. Starting from 9 dph, two experimental groups were established based on the presence or absence of newly hatched larvae: formulated diet Group (F Group)—fed only a formulated diet, and formulated diet + newly hatched larvae Group (FL Group)—fed a combination of a formulated diet and striped beakfish newly hatched larvae. Each group consisted of five replicates.

#### 2.2.3. Sampling and Proportional Measurement of Larvae and Juveniles

Each day, 30 larvae or juveniles were randomly sampled from each tank, anesthetize with eugenol (FA100, Bussan Animal Health Co., Ltd., Osaka, Japan) and then euthanized in ice water, and measured body length (BL; notochord length (NL) or standard length (SL) under a microscope (cellSense standard 1.16, OLYMPUS Co., Tokyo, Japan; SZX7, OLYMPUS Co., Tokyo, Japan).

#### 2.2.4. Method for Estimating Survival Rate

On the final day of the rearing trial, all larvae remaining in the tank were collected and the total number of larvae was counted.

#### 2.2.5. Statistical Analysis

Statistical analyses were conducted using the statistical software EZR 1.67 [45]. Differences between experimental groups were tested using analysis of variance (ANOVA), with a significance level set at 5% (*p* < 0.05). For body length (BL), a *t*-test was performed, while variance in length was assessed using an F-test. Survival rates were analyzed using the Mann–Whitney U test.

### 2.3. Experiment 2: Relative Growth from Larval to Juvenile Stages in Kawakawa

#### 2.3.1. Test Larvae and Juveniles and Their Rearing Methods

The test fish used in this experiment were obtained from fertilized eggs as the same source as Experiment 1. A total of 100,000 fertilized eggs were stocked in each of two indoor 30 m^3^ circular tanks for rearing.

During the rearing period, dissolved oxygen (DO) levels, ranged from 100% to 120%, remaining within the appropriate range. Water temperature was maintained at 26.5 °C until 4 dph, after which it was adjusted to 27.0 °C from 5 dph onward and maintained until the end of the experiment. An illuminance of 2000–5500 lx was provided, with a photoperiod of 12L:12D (light from 06:00 to 18:00).

From 2 to 12 dph, kawakawa larvae were fed rotifers. The rotifers were enriched with EPA- and DHA-fortified freshwater Chlorella. From 9 to 22 dph, in addition to rotifers, larvae were provided with striped beakfish newly hatched larvae sourced from the Kindai University Aquaculture Technology and Production Center Oshima Branch, with a feeding amount of 25 billion newly hatched larvae per tank. From 7 dph until offshore transfer, a formulated diet (Magokoro, Nisshin Marubeni Feed Co., Tokyo, Japan) was provided in appropriate amounts.

The seawater used for rearing was UV-sterilized and filtered, with an appropriate amount of concentrated *Chlorella* algae (Fresh Chlorella V-12, Chlorella Industry Co., Tokyo, Japan) added as needed.

#### 2.3.2. Sampling and Measurement of Larvae and Juveniles

Each day, 30 larvae or juveniles were randomly sampled from each tank, anesthetize with eugenol (FA100, Bussan Animal Health Co., Ltd., Osaka, Japan) and then euthanized in ice water, and digital images were taken (cellSense standard 1.16, OLYMPUS Co., Tokyo, Japan; SZX7, OLYMPUS Co., Tokyo, Japan), and the lengths of body parts were measured from the images. For absolute growth assessment, notochord length (NL) and standard length (SL) were measured and these were termed ‘body length (BL)’. For relative growth analysis, the following morphological parameters were evaluated: total length (TL), body length (BL), preanal length (PAL), head length (HL), snout length (SnL), body height (BH), head height (HH), caudal peduncle depth (CPD), upper jaw length (UJL), and eye diameter (ED). The test fish were larvae and juveniles reared for industrial-scale fingerling production at the Kindai University Aquaculture Technology and Production Center. A total of 10 to 30 individuals were randomly sampled at 1, 2, 3, 4, 5, 6, 10, 11, 13, 15, 17, 18, and 21 dph. The developmental stages of kawakawa were also referenced according to Kendall (1984) [46].

#### 2.3.3. Statistical Analysis

Statistical analysis was performed using EZR 1.67 [45]. To determine whether the slopes of all regression lines in the relative growth analysis of kawakawa larvae and juveniles significantly differed before and after the inflection points, analysis of covariance (ANCOVA) was conducted. A significance level was set at 5% (*p* < 0.05), and statistically significant differences were found in all comparisons of slopes before and after the inflection points.

## 3. Results

### 3.1. Experiment 1

#### 3.1.1. Absolute Growth

The diameter of the fertilized eggs was 0.94 ± 0.03 mm (*n* = 30), and the oil globule diameter was 0.23 ± 0.02 mm (*n* = 30). On 1 dph, larval size was 2.62 ± 0.20 mmBL, 4.27 ± 0.40 mmBL on 5 dph, and on 9 dph, when rotifers, newly hatched larvae, and formulated diet were provided, it was 5.46 ± 0.63 mmBL. In the group fed only with formulated diet, larval size was 6.13 ± 0.75 mmBL on 10 dph, 6.33 ± 0.45 mmBL on 11 dph, 6.82 ± 0.38 mmBL on 12 dph, and 7.02 ± 0.36 mmBL on 13 dph. In the group fed with both formulated diet and newly hatched larvae, larval size was 6.02 ± 0.70 mmBL on 10 dph, 6.97± 0.626 mmBL on 11 dph, 8.34 ± 1.030 mmBL on 12 dph, and 9.56 ± 0.82 mmBL on 13 dph (36% greater than that of F Group larvae). In this wise, from 12 dph, kawakawa exhibited accelerated growth and statistical analysis showed that from 12 dph onward, the mean BL and its variance were significantly greater in the formulated diet and newly hatched larvae group (FL Group) compared to the formulated diet group (F Group; *p* < 0.05, Figure 1).

#### 3.1.2. Survival Rate

The survival rate at 13 dph was 6.4% in the F Group, and 3.8% in the FL Group (Figure 2), which was 34% lower than that in F Group. The survival rate was significantly higher in the F Group compared to the FL Group (*p* < 0.05).

### 3.2. Experiment 2: Relative Growth

The absolute growth of BL in the sampled larvae and juveniles showed the following values: 3.90 ± 0.46 mm at 5 dph, 7.51 ± 0.189 mm at 10 dph, 12.72 ± 1.25 mm at 13 dph, and 47.58 ± 4.03 mm at 21 dph (Figure 2). Growth accelerated around 13 dph (Figure 3a). Additionally, growth up to 13 dph was slightly greater in larvae reared in 30 t volume tanks compared to those in 1 t volume tanks, where different feeding regimens were tested.

Relative growth patterns of each body part are shown in Figure 4 and Table 1. Growth inflection points were observed in all measured body parts, with notable inflection points concentrated at 3 mm, 10 mm, and 30 mm in BL. The PAL relative to BL changed with growth, measuring approximately 30% at 3 mmBL, 63% at 10 mmBL, and 70% at 30 mm SL, indicating a posterior shift in the position of the anus as the fish grew.

#### 3.2.1. Absolute Growth

The egg diameter and oil globule diameter were measured, with mean values of 0.73 ± 0.03 mm and 0.19 ± 0.02 mm, respectively. The mean BL and TL at different developmental stages were as follows: 1 dph: 2.75 ± 0.16 mmBL, 2.85 ± 0.18 mmTL, 2 dph: 2.91 ± 0.15 mmBL, 3.02 ± 0.16 mmTL, 5 dph: 3.85 ± 0.23 mmBL, 3.90 ± 0.46 mmTL, 8 dph: 5.44 ± 0.45 mmBL, 5.76 ± 0.46 mmTL, 10 dph: 6.85 ± 0.41 mmBL, 7.51 ± 0.19 mmTL, 18 dph: 33.39 ± 3.00 mmBL, 38.53 ± 3.36 mmTL (Figure 3).

#### 3.2.2. Development

At 2 dph, larvae transitioned from the early larval stage to the pre-flexion stage, shifting from endogenous to mixed nutrition as they began exogenous feeding (GIF image of 2 dph larvae attached). At 5 dph (mean BL: 3.89 mm), tooth development began, and notochord flexion initiated (Figure 3). By 8 dph (mean BL: 5.44 mm), most individuals had entered the flexion stage. At 10 dph (mean BL: 6.85 mm), all individuals had reached the late flexion stage. Rotifer feeding ceased at this stage, and the diet transitioned to formulated feed and newly hatched larvae. At 18 dph (mean BL: 33.39 mm), the number of fin rays reached their species-specific counts, and all individuals had developed into the juvenile stage (Figure 5).

#### 3.2.3. Relative Growth of Upper Jaw Length at the Onset of Newly Hatched Larvae Feeding

At 10 dph, when newly hatched larvae of Japanese beakperch feeding began, the mean BL was 6.9 mm, and the mean UJL was 1.8 mm.

## 4. Discussion

### 4.1. Effects of Newly Hatched Larvae Feeding on Larval Growth and Survival of Kawakawa

#### 4.1.1. Effect of Larval Feeding on Growth Acceleration

In this study, kawakawa exhibited accelerated growth around 13 dph, when body length reached approximately 10 mm (12 mmTL) at flexion stage (Figure 3). In comparison, growth acceleration in Pacific bluefin tuna occurred around 20 dph at 18 mmTL [15], in yellowfin tuna around 22 dph at 17 mmTL [20], and in chub mackerel (*Scomber japonicus*) around 10 dph at 6 mmTL (postflexion) [44]. These growth acceleration points coincide with growth inflection points identified in the relative growth patterns of each species as discussed in the later sections.

During the first 8 days, no growth difference was observed between the two groups fed only rotifers. However, after 9 dph, feeding newly hatched larvae led to significantly greater growth in the FL group, indicating its promotive effect on kawakawa larval growth.

Docosahexaenoic acid (DHA) has been reported to promote growth and improve survival in fish larvae. *Artemia* nauplii, which are commonly used as feed in the larval rearing of many marine fish species, are not used in the larval rearing of kawakawa. This may be due to the inferior nutritional characteristics of *Artemia* nauplii compared to hatched larvae, as described below. Previous studies have analyzed the nutrient composition of various larval feeds, including enriched rotifers, *Artemia* nauplii, and striped beakfish newly hatched larvae, and examined their effects on larval growth and body composition in Pacific bluefin tuna [47]. The study found that striped beakfish larvae and enriched rotifers contained 23 times higher DHA content in the polar lipid fraction compared to enriched *Artemia* nauplii. Larvae fed *Artemia* nauplii exhibited inferior growth compared to those fed striped beakfish larvae, and the DHA content in larvae was only one-third that of the latter group. Subsequent research by Seoka et al. (2008) demonstrated that enhancing the DHA content of *Artemia* nauplii led to improved growth and survival in Pacific bluefin tuna larvae [48]. Similarly, in Atlantic bluefin tuna, feeding larvae yolk-sac larvae of gilthead sea bream (*Sparus aurata*) resulted in increased. DHA content and enhanced growth [49]. Numerous studies have reported that marine fish larvae require DHA for optimal growth and normal development [50,51]. In scombrid species, early developmental stages are particularly dependent on DHA, as demonstrated in Pacific bluefin tuna [47,48,49,50,51,52,53,54] and in Atlantic bluefin tuna [32,54,55]. These findings highlight the critical role of DHA in the early growth and survival of scombrid larvae, further supporting the observed benefits of hatched larvae fish feeding in kawakawa larvae. Dietary DHA plays important roles in osmoregulation, bile salt conjugation, membrane stabilization, modulation of neurotransmitters, antioxidation and development of visual. In kawakawa, further nutritional studies are needed to evaluate the effects of hatched larvae fish feeding. This should include an analysis of feed composition and larval body composition, particularly focusing on DHA content, to better understand its impact on growth and survival.

#### 4.1.2. Poor Survival

Cannibalism is another possible factor contributing to the growth that appeared to have accelerated at first glance of kawakawa larvae fed striped beakfish newly hatched larvae, which emerges early due to the species’ strong piscivorous tendencies. At 13 dph, the survival rate in F Group was 6.36 ± 1.43%, which was significantly higher than the 3.83 ± 1.36% observed in the FL Group (*p* < 0.05; Figure 2). This suggests that size variation among individuals increased when hatched larvae were included in the diet, as some larvae grew faster while others did not. Consequently, larger individuals likely preyed on smaller ones, a pattern commonly observed in other scombrid species, such as Pacific bluefin tuna [31], yellowfin tuna [56], and striped bonito [22], and this might result in the survival of large-size larvae, which made it appear as if growth had accelerated.

Kawakawa may develop piscivory earlier than yellowfin and Pacific bluefin tuna, allowing them to consume larger prey at an earlier stage, which may enable rapid acceleration of growth. In other words, the enhanced growth observed in the FL group may be explained not only by the nutritional advantages conferred by feeding with hatched larvae, but also by increased size heterogeneity within the cohort, which likely induced cannibalistic behavior. As a result, larger and faster-growing individuals were more likely to survive, thereby contributing to an apparent acceleration in overall growth compared to the F group.

Importantly, environmental deterioration and disease-related factors are unlikely to account for the reduced survival observed in the FL group. Throughout the experimental period, key water quality parameters, including dissolved oxygen (DO) and pH, remained within optimal ranges, and no external signs of disease, abnormal behavior, or mass mortality events attributable to pathogenic infections were detected during daily monitoring. These observations strongly suggest that cannibalism, rather than environmental or pathological stressors, was the primary factor contributing to the lower survival in the FL group.

Cannibalism in larval and juvenile fish under captive conditions is influenced by biotic and abiotic factors, and several mitigation strategies have been proposed. Key factors affecting cannibalistic behavior include size variation among individuals [39,57], stocking density [36], availability of feeds [39,58], light conditions [59,60,61], water turbidity [62,63], water flow [64], temperature-induced size variation [65]. Regarding the biotic factors, the gastric development is prerequisite for piscivory. In the present study, we observed that kawakawa larvae began feeding on newly hatched striped beakfish at 9 dph, when they were approximately 6 mm TL and had reached the flexion stage. This timing appears to coincide with the initial development of gastric function, as reported by Duy Khoa et al. (2021) [66], who noted the onset of pepsin activity around 10 dph. This suggests that the development of the stomach and gastric glands is a key prerequisite for the initiation of piscivory in kawakawa larvae. The ability to digest complex prey such as fish larvae likely contributed to the rapid growth observed in the treatment group.

Previous studies in related species, including *Thunnus orientalis*, *T. thynnus*, *T. albacares,* and *Scomberomorus niphonius*, also support the notion that the onset of pepsin activity and gastric gland formation marks a physiological shift toward piscivory [13,14,20,67,68]. Our findings, while not directly assessing histology or enzyme activity, align with these developmental timelines and suggest that kawakawa may exhibit relatively early gastric development compared to other scombrids, which may underlie their early transition to piscivory and associated growth acceleration Yufera et al. (2014) [68] also suggested that the stomach seemed to begin functioning perfectly once the gastric glands proliferated and pepsin expression and activity progressed rapidly. In Spanish mackerel, gastric glands and pyloric caeca are already developed by 2 dph, when larvae begin feeding [67]. During larval rearing, hatched larvae fish such as red seabream (*Pagrus major*) are provided as prey from the onset of exogenous feeding [69,70]. Similarly, in striped bonito, Harada et al. (1974) [24] reported that at 10 dph (14.4 mmTL), larvae exhibited piscivorous behavior. Kaji et al. (2002) [22] found that in 3 dph pre-flexion larvae, gastric glands and pyloric caeca were already differentiated, indicating the ability to consume hatched larvae fish at this early stage.

Another important biotic factor related to piscivory, and cannibalism is size variation in the cohort of larvae and juveniles. At 17 dph, when kawakawa reached 29 mmBL (33 mmTL), size variation among individuals increased (CV: 0.08 at 15 dph → 0.16 at 17 dph). Ishibashi et al. (2014) [39] demonstrated that cannibalism in Pacific bluefin tuna larvae was primarily driven by size variation and feed shortages. Similarly, practical experience from kawakawa hatchery production suggests that abundant newly hatched larvae feeding can reduce cannibalism. However, Manabe (2019) [33] reported that raising a single kawakawa juvenile (34–42 mmTL) requires approximately 13,000 hatched larvae of Japanese seabass (*Lateolabrax japonicus*). Given that scombrid larvae require large numbers of newly hatched larvae daily (tens to hundreds per individual), providing sufficient hatched larvae fish at an industrial-scale seed production facility remains a significant challenge. Therefore, future research should focus on optimizing larvae feeding quantities, balancing growth enhancement and survival rate improvements to maximize production efficiency. To mitigate cannibalism in kawakawa larvae, environmental modifications should be explored, such as adjusting microalgae concentration, modifying light intensity, and changing tank wall color. These factors could reduce visual recognition between individuals, potentially decreasing cannibalism, and warrant further investigation. Further research is required to determine the optimal timing and feeding amount of newly hatched fish larvae for kawakawa larvae and juveniles, in order to better understand and enhance their growth performance.

### 4.2. Relative Growth of Kawakawa Larvae and Juveniles

The following discussion on inter-species comparisons of growth patterns in scombrid species references studies on Pacific bluefin tuna [15], yellowfin tuna [20], striped bonito [15], and chub mackerel [44]. This section aims to clarify the characteristics of relative growth in kawakawa larvae and juveniles, in relation to their absolute growth. It highlights that larger scombrid species, such as tunas, tend to prioritize growth over development compared to smaller scombrids like kawakawa. Furthermore, even among scombrids, relative growth patterns vary, and based on the relationship between mouthpart development and piscivory—a key feature of scombrid fishes—these species can be categorized into three distinct groups. The following discussion on inter-species comparisons of growth patterns in scombrid species references studies on Pacific bluefin tuna [15], yellowfin tuna [20], striped bonito [15], and chub mackerel [44].

#### 4.2.1. Relationship Between Absolute and Relative Growth

In kawakawa, growth acceleration occurs around 13 dph, at approximately 10 mmBL (12 mmTL). In comparison, for Pacific bluefin tuna, growth accelerates at 20 dph, at 18 mmTL, for yellowfin tuna, growth accelerates at 22 dph, at 17 mmTL, and for chub mackerel, growth accelerates at 10 dph, at 6 mmTL. These growth acceleration points align with the inflection points in relative growth patterns for each species. For species with smaller maximum sizes (striped bonito, kawakawa, and chub mackerel), isometric growth is observed in many body parts when body length exceeds 10 mm. In contrast, species with larger maximum sizes (Pacific bluefin tuna and yellowfin tuna) tend to exhibit negative allometric growth in many body parts after the same stage. This suggests that larger species prioritize increasing body size beyond the early growth inflection point, which may be essential for their ecological strategy and survival.

#### 4.2.2. Changes in Body Shape

In other scombrid species as well, the timing of inflection points and growth patterns vary depending on the species. In terms of changes in whole-body proportions, yellowfin tuna exhibit rapid positive allometric growth in several body parts until 10 mmTL, followed by a transition to slower relative growth [71]. During the rapid growth phase, the regression slopes were 0.50 for head length (HL), 0.30 for lower jaw length (LJL), 0.15 for eye diameter (ED), 0.60 for preanal length (PAL), 0.35 for body depth (BD), 0.18 for standard length (SL), and 0.30 for upper jaw length (UJL). In the subsequent slower growth phase, the slopes decreased to 0.35 (HL), 0.15 (LJL), 0.10 (ED), 0.08 (SL), 0.23 (BD), and 0.18 (UJL). This growth pattern is similar to that of kawakawa, although the regression slopes during the observed growth period are generally smaller in yellowfin tuna than in kawakawa.

In Pacific bluefin tuna, preanal length initially exhibits negative allometric growth, with a slope of −0.08, followed by a shift to positive allometry with a slope of 0.72. This transition pattern is also similar to that observed in kawakawa [15].

However, at 10 mmTL, the regression slope in Pacific bluefin tuna exceeds that of kawakawa. As growth progresses, Pacific bluefin tuna surpass kawakawa in body size and exhibit a substantial increase in positive allometry [15]. In kawakawa, the regression slopes during the rapid growth phase are 1.19 for total length (TL), 0.72 for PAL, 0.50 for HL, 0.25 for SL, 0.36 for body height (BH), 0.40 for head height (HH), 0.10 for caudal peduncle depth (CPD), 0.40 for UJL, and 0.14 for ED. In the slower growth phase, the slopes were 1.18 (TL), 0.75 (PAL), 0.34 (HL), 0.13 (SL), 0.15 (BH), 0.20 (HH), 0.01 (CPD), 0.16 (UJL), and 0.07 (ED), indicating a distinct deceleration in relative growth.

In chub mackerel, an inflection point was observed at approximately 5 mmBL, after which the growth shifted to a relatively gradual positive allometry. An inflection point was seen at approximately 10 mmBL after which isometric growth was observed for all body parts, [44]. These observations indicate that the inflection points in body length and the corresponding relative growth patterns differ among scombrid species.

In kawakawa, three distinct inflection points were observed for multiple body parts (total length, preanal length, head length, snout length, body height, and caudal peduncle depth) during the development. At 3.85 mmBL (3.90 mmTL), preanal length exhibited a steep negative allometric growth with a regression slope of −0.11. The degree of negative growth has decreased (slope: 0.04 to 0.01) for larger body lengths, at 10 mmBL (12 mmTL), the most significant change in growth occurred. After this inflection point, total length and preanal length exhibited negative allometric growth (slopes of 0 and 0.01, respectively), while other body parts shifted to negative allometric growth, and at 30 mmBL (35 mmTL), most body parts showed negative allometric growth prior to the inflection point but transitioned to negative allometric growth (slope: 1.00) beyond this point for preanal length.

The overall patterns of relative growth in kawakawa were evaluated in comparison with those observed in other scombrid fishes. Pacific bluefin tuna exhibited a different growth pattern at: approximately 5–6 mmTL, both total length and preanal length transitioned from negative to positive allometric growth, 10 mmTL, most other body parts shifted to negative allometric growth [15]. Cultured yellowfin tuna exhibited inflection points at: 10 mmTL for head height, snout length, body height, upper jaw length, and caudal peduncle depth, 15 mmTL for eye diameter and preanal length, with a transition from positive to negative allometric growth. Yellowfin tuna also had an additional inflection point at 5 mmTL for body height [71]. In striped bonito, an inflection point at 6 mmTL was followed by rapid positive growth for all body parts, between 8 and 13 mmTL, total length, eye diameter, snout length, upper jaw length, and head length transitioned to isometric growth, and preanal length transitioned at 30 mmTL, showing isometric growth thereafter [25]. In chub mackerel, inflection points were observed at: 5 mmTL for total length and body height, followed by gentle positive allometric growth, and 10 mmTL for all body parts, with subsequent isometric growth [44].

#### 4.2.3. Evaluation of Newly Hatched Larvae Feeding as the Feeding Strategies in Scombrid Fingerling Production Based on Their Relative Growth from the View Point of Their Strong Piscivory

A comparison of upper jaw growth among scombrid species reveals the following ranking in terms of the slopes of positive allometric equations of upper jaw length relative to body length: striped bonito (15.0) > kawakawa (3.8) > yellowfin tuna (3.6) > Pacific bluefin tuna (2.3) > chub mackerel (1.2). Scombrid larvae are known for their large mouth openings, a characteristic reported in multiple species [72,73]. Upper jaw length is a critical indicator of fish feeding capability [74,75]. Table 2 summarizes the upper jaw length (UJL) relative to body length (BL) at the BL which starts growth acceleration in each scombrid species. In terms of body size at the point of growth acceleration: in kawakawa growth accelerates at 13 dph (11.2 mmBL), Pacific bluefin tuna at 20 dph (18 mmBL), yellowfin tuna at 22 dph (17 mmBL), and chub mackerel at 10 dph (6 mmBL). For most of these species (except for chub mackerel), growth acceleration corresponds with the introduction of newly hatched larvae as feed [15,20,25]. On the other hand, these results indicate that upper jaw growth and relative body proportions differ significantly among scombrid species, reflecting their unique ecological roles and feeding strategies. Kawakawa and striped bonito exhibit rapid early growth of feeding-related structures, which may support their early piscivory and fast growth rates. At the body length where jaw growth accelerates, the relative upper jaw length (UJL/BL) is 26.8% in kawakawa (11.2 mmBL) and 25.0% in striped bonito (8.0–10.0 mmBL). In contrast, Pacific bluefin tuna and yellowfin tuna prioritize overall body growth during later stages, with UJL/BL values of 20.0% at 18 mm and 17 mmBL, respectively. Chub mackerel show a more gradual growth trajectory with less dramatic changes in feeding structures, and a lower relative upper jaw length of 13.3% at 6 mmBL. Striped bonito hatch at approximately 3.6 mmTL, which is considerably larger than kawakawa at hatching (2.75 ± 0.16 mmTL).

In the current kawakawa aquaculture protocol, newly hatched striped beakfish larvae are introduced as feed when larvae reach 6 mmBL, at which point their upper jaw length is approximately 1.54 mm (25% BL). Similarly, in Pacific bluefin tuna and yellowfin tuna, newly hatched larvae are introduced when the larvae reach 8 mmTL, with the following upper jaw measurements: Pacific bluefin tuna: 1.44 mmUJL (17% BL), yellowfin tuna: 1.84 mmUJL (23% BL). In contrast, chub mackerel have a much smaller upper jaw relative to body length, with the proportion of UJL to notochord length ranging from 7 to 10% throughout the measured period (3.1 mm to 90.9 mmBL, [44]). Chub mackerel larvae are not fed newly hatched larvae during rearing [44]. Although adult chub mackerel are piscivorous [76], their small upper jaw size during early stages likely delays the onset of piscivory, making them unsuitable for large prey at the larval stage.

Based on the relative upper jaw size (Table 2) and feeding protocols in this study and those in previous studies, scombrid species can be classified into three distinct groups.

The regression slopes of upper jaw length to body length until the first flexion point were 0.39 in kawakawa, 0.27 in striped bonito, 0.40 in yellowfin tuna, 0.27 in Pacific bluefin tuna, and 0.18 in chub mackerel. These values quantitatively support the grouping based on relative upper jaw size. In Group I, which includes striped bonito and bullet tuna, their larvae are provided with the newly hatched larvae of other fishes from the early stage of mouth opening. Group II consists of species with a relatively larger upper jaw, such as yellowfin tuna and Pacific bluefin tuna. In these species, larvae are fed with the newly hatched larvae of other fish species only after the postflexion stage. Group III includes species with a relatively smaller upper jaw, such as chub mackerel. In this species, larvae are not provided with the newly hatched larvae of other fishes as food. This classification is similar to the previous study by Richards (1973) [77]. He proposed that certain characteristics of larval fish may suggest intergeneric relationships among scombrids in association with myomere counts. His classification is as follows. Group I consists of Scomber (mackerel), which has a low number of myomeres (30–31). Group II includes Auxis (bullet tuna), Katsuwonus (skipjack tuna), Euthynnus (little tunny), Thunnus (tuna), and Allothunnus (slender tuna), which have a moderate number of myomeres (38–43) and a large head. Group III comprises Sarda (bonito), Scomberomorus (Spanish mackerel), and Acanthocybium (wahoo), which have a high number of myomeres (43–63) and a progressively increasing snout length. Although the criteria used for classification are different between the studies, the results that scombrids are divided into three groups are consistent. Additionally, Matsumoto et al. (1967) stated that the size of the premaxillary representing the upper jaw size in the larval stage may be of generic significance in the taxonomy of scombrids [78]. The upper jaw size was used as the criteria of scombrid taxonomy in both these studies.

## 5. Conclusions

This study suggests that hatched larvae feeding benefits kawakawa larvae by promoting growth, although it may also intensify size variation and cannibalism. At present, the feeding of live fish larvae from other fish species to kawakawa larvae is essential for its fingerling production, as there are no superior alternative formulated feeds available and the approach is biomimetic in nature. Given the high DHA requirements of scombrid larvae, future research should focus on optimizing hatched larval feeding quantities to balance growth and survival, exploring environmental modifications to mitigate cannibalism, developing alternative artificial diets enriched in DHA and other nutrients. By addressing these challenges, more efficient large-scale fingerling production of kawakawa can be achieved.

## Figures and Tables

**Figure 1 animals-15-01997-f001:**
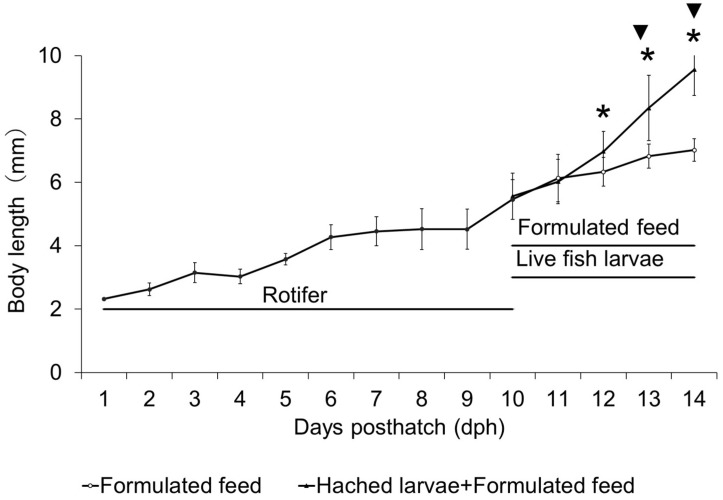
Absolute growth of body length (notochord length or standard length) of kawakawa larvae and juveniles, and feeding regime in Experiment 1. Vertical lines indicate standard deviations (*n* = 30). * and ▼ denote significant differences in the means and variances, respectively (*p* < 0.05).

**Figure 2 animals-15-01997-f002:**
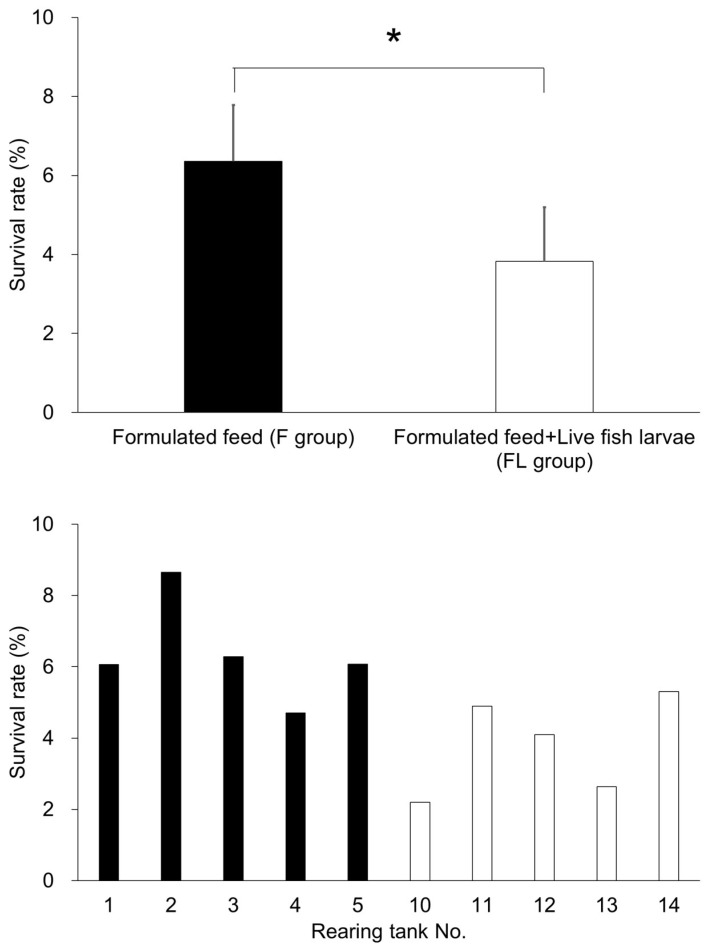
Survival rates in the formulated diet group (F group) and the formulated diet plus live fish group (FL group). The upper panel shows the average survival rate for each experimental group, while the lower panel presents the survival rates in individual rearing tanks (five replicates per group). * indicates the significant difference (*p* < 0.05).

**Figure 3 animals-15-01997-f003:**
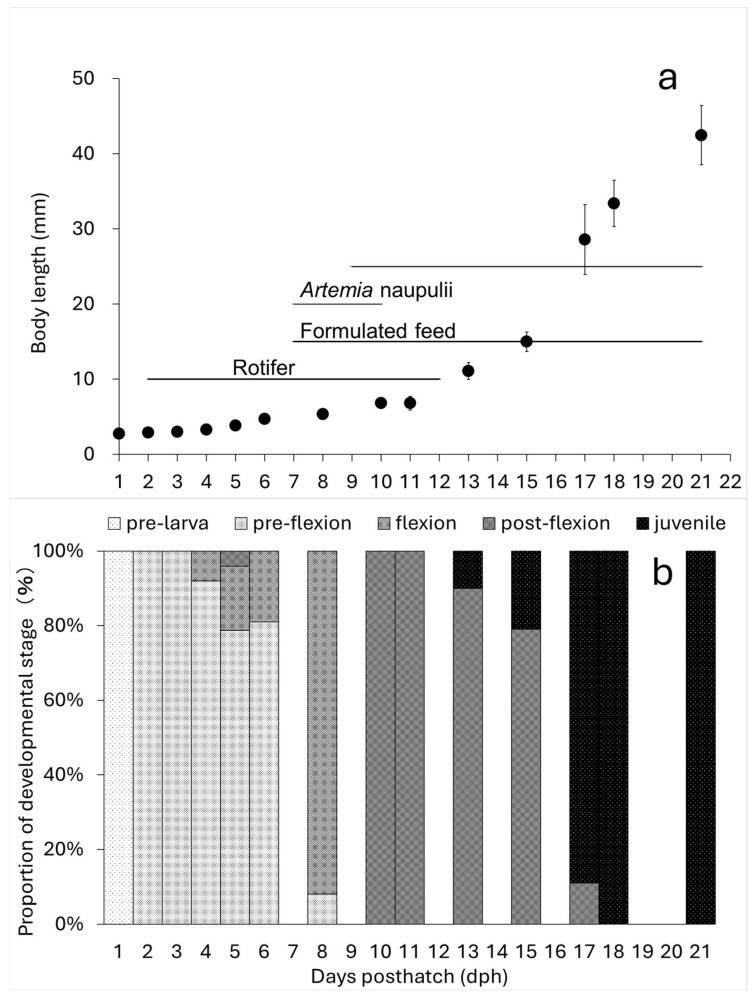
Absolute growth of body length (notochord length or standard length) of kawakawa larvae and juveniles, their feeding regime, (upper panel; (**a**)) and developmental stage (lower panel; (**b**)) in Experiment 2. Vertical lines indicate ±SD (*n* = 15–30).

**Figure 4 animals-15-01997-f004:**
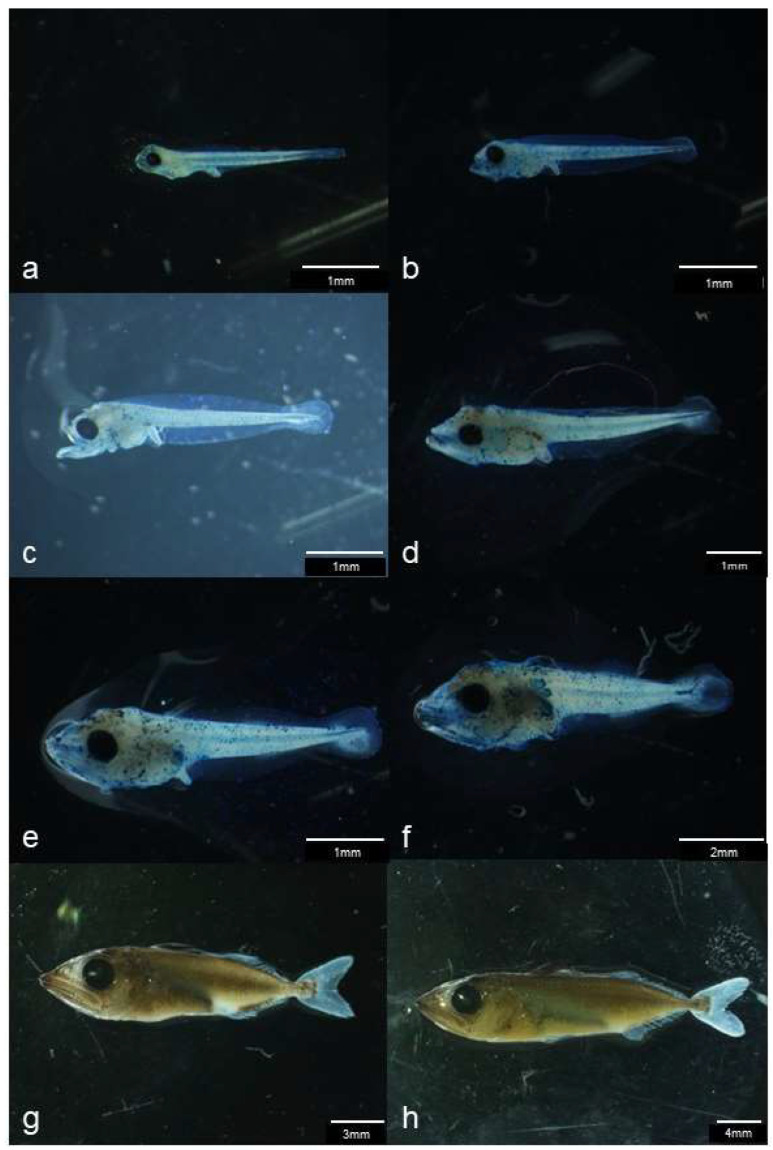
Images of kawakawa (*Euthynnus affinis*); (**a**): 1-day post-hatch (dph) larva, (**b**): 3 dph individual approximately 3 mm in body length, (**c**): 4 dph pre-flexion stage larva, (**d**): 6 dph flexion stage larva, (**e**): 8 dph post-flexion stage larva, (**f**): 10 dph individual approximately 9 mm in body length, (**g**): 15 dph juvenile, (**h**): 17 dph individual approximately 30 mm in body length.

**Figure 5 animals-15-01997-f005:**
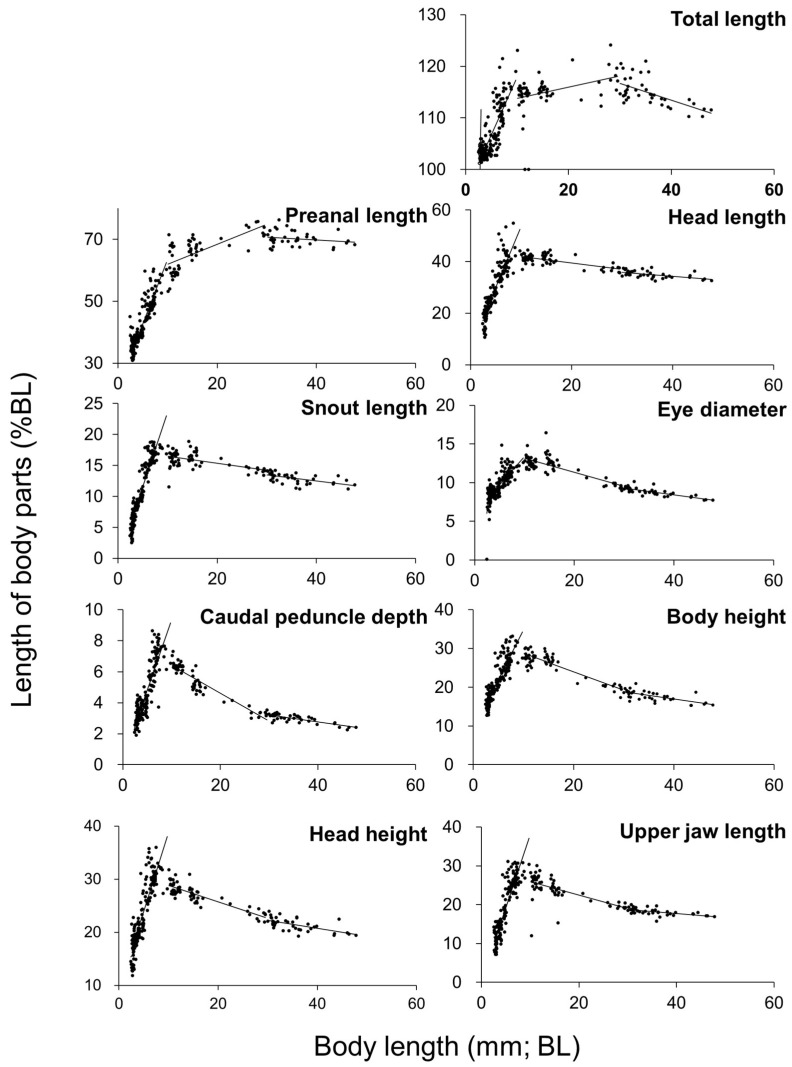
Relative growth of kawakawa larval and juvenile body parts in Experiment 2, shown as a percentage of body length (notochord length or standard length).

**Table 1 animals-15-01997-t001:** Allometry of body part lengths in kawakawa larvae and juveniles.

	Allometry (*y* = a + b*x*; *y*: Length Body Parts, *x*: Body Length)			
Body Part	Range of Body Length	a	b	Correlation Coefficient (R^2^)
Total length	BL < 3	−30.43	114.54	0.97
	3 ≤ BL < 10	−55.48	119.12	0.99
	10 ≤ BL < 30	−38.49	117.67	0.99
Preanal length	BL < 3	63.23	−9.70	0.27
	3 ≤ BL < 10	22.02	4.15	0.82
	10 ≤ BL < 30	56.78	0.55	0.44
	30 ≤ BL	75.11	0.13	0.09
Head length	3 ≤ BL < 10	7.00	4.67	0.80
	10 ≤ BL < 30	44.65	−0.27	0.80
	30 ≤ BL	41.01	−0.17	0.37
Snout length	3 ≤ BL < 10	−0.90	2.50	0.89
	10 ≤ BL < 30	17.75	−0.12	0.31
	30 ≤ BL	16.62	−0.10	0.40
Body height	BL < 3	24.98	−3.73	0.19
	3 ≤ BL < 10	9.04	2.62	0.79
	10 ≤ BL < 30	33.05	−0.44	0.79
Head height	3 ≤ BL < 10	9.37	2.95	0.78
	10 ≤ BL < 30	31.64	−0.29	0.75
	30 ≤ BL	26.46	−0.14	0.40
Caudal peduncle depth	BL < 3	1.49	0.43	0.03
	3 ≤ BL < 10	0.41	0.90	0.74
	10 ≤ BL < 30	8.08	−0.17	0.87
	30 ≤ BL	4.54	−0.04	0.67
Upper jaw length	10 ≤ BL < 30	29.45	−0.35	0.42
	30 < BL	22.09	−0.11	0.40
Eye diameter	3 ≤ BL < 10	6.08	0.73	0.63
	10 ≤ BL < 30	15.08	−0.19	0.65

**Table 2 animals-15-01997-t002:** Upper jaw length relative to body length at the growth acceleration.

Species	Body Length (mm)	Upper Jaw Length (mm)	UJL/BL (%)	Reference
Chub mackerel	6.0	0.8	13.3	473
Striped bonito	8.0–10.0	2.5	25.0	25
Kawakawa	11.2	3.0	26.8	This study
Yellowfin tuna	17.0	3.4	20.0	74
Pacific bluefin tuna	18.0	3.6	20.0	15

## Data Availability

Data are contained within the article.

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
