# Peer review of "Effects of Feeding Newly Hatched Larvae on the Growth, Survival, and Growth Patterns of Kawakawa (Euthynnus affinis) Larvae and Juveniles"

_animals, 2025, doi:10.3390/ani15131997_

Round 1
Reviewer 1 Report
Comments and Suggestions for Authors
The manuscript may be reconsidered after major revision.
The authors state that the optimal timing and quantity for feeding hatched kawakawa larvae remain unclear; however, this experiment does not address this issue.
Why did the authors not design the experiment to examine different timings and quantities of striped beakfish?
Additionally, the rationale for selecting striped beakfish as live prey for kawakawa is not clearly explained. Is it feasible to produce this species at a scale sufficient to support kawakawa aquaculture, and does its use represent a reasonable economic cost for commercial application?
The Discussion should focus on this study’s results comparing control and treatment groups, rather than on other scombrid species.
The Discussion does not align with the experimental design and results.
Abstract
Line 24-26; The authors state that 10,000 striped beakfish larvae were used during the larval stage, while a different amount appears to have been used during the juvenile stage. This may cause confusion for readers. I recommend clarifying this point to ensure consistency and avoid misunderstanding.
Line 28; However, survival at 13 dph was approximately 34% lower.
Could this be attributed to water quality issues, the presence of pathogens, feeding technique or other environmental factors? It would be valuable if the authors could clarify whether any investigation was conducted to identify the underlying cause.
Line 25; they were assigned to two experimental groups: one receiving only a formulated diet and the other receiving a combination of the formulated diet and 10,000 striped beakfish newly-hatched larvae every day in each tank.
Could the authors clarify why this specific concentration was selected and why varying feeding levels were not tested?
Introduction
Line 70; In kawakawa, strong piscivorous behavior, which has been reported on around 12 dph.
The authors initiated the feeding experiment between day 9 and day 12 post-hatching, despite reports indicating the onset of strong piscivorous behavior in kawakawa around 12 dph. Could the authors clarify the rationale for selecting this specific timeframe and explain why a longer feeding period was not considered?
Could the authors clarify the reason for selecting striped beakfish (Oplegnathus fasciatus) as live prey for kawakawa? Is it feasible to produce this species at a scale sufficient to support kawakawa aquaculture, and does its use present a reasonable economic cost for commercial application? This information should be included in the introduction.
Materials and methods
The authors stated that digital images were taken and used to measure body length (Lines 167, 211, 221); however, no images are provided in the manuscript. I recommend including these images as supplementary materials to enhance clarity and transparency for readers.
Please reduce the redundant information in Sections 2.2 and 2.3.
The authors seem to indicate that different amounts of striped beakfish were used in Sections 2.2 and 2.3. It is unclear why the same feeding protocol and management were not applied in both experiments.
Results
In Section 3.1.2 (Survival Rate), the authors present both the average survival rate across all replicate tanks and data from individual tanks, shown in two separate graphs. However, the individual tank data is not clearly described. The reason for presenting the data separately should be clarified. In addition, the figure legend does not adequately describe the two graphs. A clearer explanation is needed to help readers interpret the data.
The titles of Sections 3.1 and 3.2 do not clearly reflect the specific focus of this experiment. Revising the section headings to better align with the content would improve clarity.
Please revise the order of presentation so that tables and figures appear in the sequence in which they are cited in the text—for example, Table 1 is placed before Figure 3, but Figure 3 is mentioned earlier in the manuscript.
The authors should separate Figure 3 into Figures 3a and 3b and revise the corresponding references in the manuscript text accordingly.
Section 3.2.3 lacks a description of the results presented in Table 2. Please include relevant text to interpret and explain these findings.
Discussion
Please avoid citing figure and table numbers in the Discussion. Focus on interpretation rather than result description.
Lines 313–322 repeat results. Please revise to focus on interpretation and review the entire Discussion.
Why do Lines 232–247 focus on increased DHA and growth when the study aims to evaluate the effect of striped beakfish supplementation on kawakawa growth? Please clarify this discrepancy. The manuscript title highlights feeding newly-hatched larvae, so the content should align with this focus. If discussing DHA, the authors should provide details on DHA content in the diet or proximate composition of the fish after feeding Kawakawa.
Line 353-355 “This suggests that size variation among individuals increased when hatched larvae were included in the diet, as some larvae grew faster while others did not.”
How did the authors ensure this size difference was caused by the diet rather than inconsistencies or poor feeding technique? Please clarify.
The sentence in Lines [355-358] is grammatically unclear and needs revision for clarity. For example, the phrase “which made it appear as if growth had accelerated. hatched larvae” is incomplete and confusing. Please rewrite for better flow and accuracy.
Section 4.2, “Relative growth of kawakawa larvae and juveniles,” is unclear. Please clarify the main points and improve the explanation for better understanding.
The Discussion should focus on interpreting the results from this study, specifically comparing the effects of feeding live feed between the control and treatment groups. However, it currently emphasizes comparisons with other scombrid larvae. Please revise to prioritize your own study’s findings.
All scientific names should be italicized throughout the manuscript. Please check for consistency.
Comments on the Quality of English LanguageThe manuscript is generally understandable; however, some sentences require revision for grammar, clarity, and flow.
Author Response
Comments 1: [The authors state that the optimal timing and quantity for feeding hatched kawakawa larvae remain unclear; however, this experiment does not address this issue. Why did the authors not design the experiment to examine different timings and quantities of striped beakfish?]
Response 1: [The primary objective of this study was to evaluate the effect of administering live hatchling larvae of other fish species as an initial feed to Euthynnus affinis larvae and juveniles.
As an initial step, we focused on assessing the basic effectiveness of this feeding method by providing a small amount of hatchling larvae at a time empirically determined to be the most likely for active ingestion by E. affinis larvae. The quantity was carefully controlled so that no noticeable feed remained the following day.
Based on the outcome of this first step, future studies can be designed to further examine the optimal timing and feeding quantity of hatchling larvae.
This paper reports the results of the first step—evaluating the effect of hatchling larvae feeding itself. The data obtained here serve as the basis for developing the next phase of investigation, which will focus on refining the timing and amount of larval fish feeding. These elements will be addressed in subsequent studies as part of a stepwise research strategy.]
Comments 2: [Additionally, the rationale for selecting striped beakfish as live prey for kawakawa is not clearly explained. Is it feasible to produce this species at a scale sufficient to support kawakawa aquaculture, and does its use represent a reasonable economic cost for commercial application?]
Response 2: [We appreciate the reviewer’s insightful comment. In response, we have added a detailed explanation in the Discussion section to clarify the rationale for selecting striped beakfish (Oplegnathus fasciatus) as live prey for kawakawa (Euthynnus affinis).
Specifically, the spawning season of striped beakfish overlaps with that of kawakawa, which makes it feasible to obtain hatchling larvae naturally during the period when kawakawa larvae require initial feeding. As a result, there is no need for artificial control of seawater temperature (such as heating or cooling) to induce spawning in striped beakfish, leading to reduced broodstock maintenance costs.
Furthermore, striped beakfish are highly fecund, enabling the mass production of fertilized eggs and hatched larvae. These biological and practical features support the potential of striped beakfish as a cost-effective and scalable live prey source for kawakawa aquaculture.
This explanation has been incorporated into the revised manuscript (Discussion section, page 11, lines ).]
Comments 3: [The Discussion should focus on this study’s results comparing control and treatment groups, rather than on other scombrid species. The Discussion does not align with the experimental design and results.]
Response 3: [
We thank the reviewer for this comment. The present study comprises two major components:
(1) characterization of the relative growth patterns in Euthynnus affinis larvae, and
(2) evaluation of the effects of feeding live hatchling larvae of other fish species as an initial diet.
Regarding the first component, the comparison with other scombrid species is essential to describe and interpret the observed relative growth patterns of E. affinis larvae. This comparative context not only facilitates a better understanding of species-specific traits in E. affinis, but also contributes to a broader understanding of relative growth trends within the Scombridae family. Based on this rationale, we included relevant comparisons in the Discussion section.
For the second component—evaluation of the effects of live hatchling prey—we have limited the comparisons to a minimum, mentioning only that similar feeding approaches have been applied in other scombrid species. Apart from this contextual reference, the focus remains strictly on the comparison between the control and treatment groups within the present study.
Therefore, we believe the Discussion is aligned with the experimental design and results, while providing appropriate context where necessary. No major structural changes were made to this section, but we remain open to further revision should the reviewer recommend more specific adjustments.]
Comments 4: [Line 24-26; The authors state that 10,000 striped beakfish larvae were used during the larval stage, while a different amount appears to have been used during the juvenile stage. This may cause confusion for readers. I recommend clarifying this point to ensure consistency and avoid misunderstanding.]
Response 4: [We did not change the feeding amount of the hatched larvae between the larval and juvenile stages. To clarify for the readers that we consistently provided 10,000 hatched larvae per tank, we have added this information in another part of the manuscript as follows.
4.1.1. Effect of larval feeding on growth acceleration
In this study, kawakawa exhibited accelerated growth around 13 dph, when body length reached approximately 10 mm (12 mmTL) at flexion stage (Fig. 3). In comparison, growth acceleration in Pacific bluefin tuna (Thunnus orientalis) occurred around 20 dph at 18 mmTL [15], in yellowfin tuna (Thunnus albacares) around 22 dph at 17 mmTL [46], and in chub mackerel (Scomber japonicus) around 10 dph at 6 mmTL (postflexion) [47]. These growth acceleration points coincide with growth inflection points identified in the relative growth patterns of each species as discussed in the later sections.
During the first 8 dph days in this study, when all larvae were fed only rotifers, no significant difference in BL was observed between the two experimental groups: formulated diet group (F Group; fed only formulated feed from 9 dph onward), formulated diet and newly-hatched larvae group (FL Group; fed formulated feed plus 10, 000 striped beakfish, Oplegnathus fasciatus, hatched larvae once every morning from 9 dph onward).]
Comments 5: [Line 28; However, survival at 13 dph was approximately 34% lower. Could this be attributed to water quality issues, the presence of pathogens, feeding technique or other environmental factors? It would be valuable if the authors could clarify whether any investigation was conducted to identify the underlying cause.]
Response 5: [We revised the sentence in lines 128–132 of section 2.2.1 as follows to clarify that there were no issues with the rearing environment and that no signs of infectious diseases were observed.
During the rearing period, water quality was measured daily at 08:00 and 13:00. The dis- 128 solved oxygen (DO) levels measured using a DO meter (550A, YSI, Maryland), ranged 129 from 83.6% to 107.6%. The pH values (D-24, HORIBA, Kyoto) were between 8.1 and 8.4, 130 while salinity levels (EC300, YSI, Maryland) ranged from 31.3 to 32.4 ppt, all within ap- 131 propriate ranges.
↓
2.2.1 lines 126-131
During the rearing period, water quality was measured daily at 08:00 and 13:00. During the experiment, the dissolved oxygen (DO) level ranged from 83.6% to 107.6%. The pH values were between 8.1 and 8.4, while salinity levels (EC300, YSI, Maryland) ranged from 31.3 to 32.4 ppt, all within appropriate ranges. Furthermore, throughout the experiment, the kawakawa larvae exhibited no signs of disease.
In addition, we added the following sentence to lines 368–375 in section 4.1.2.
Importantly, environmental deterioration and disease-related factors are unlikely to account for the reduced survival observed in the FL group. Throughout the experimental period, key water quality parameters, including dissolved oxygen (DO) and pH, remained within optimal ranges, and no external signs of disease, abnormal behavior, or mass mortality events attributable to pathogenic infections were detected during daily monitoring. These observations strongly suggest that cannibalism, rather than environmental or pathological stressors, was the primary factor contributing to the lower survival in the FL group.]
Comments 6: [Line 25; they were assigned to two experimental groups: one receiving only a formulated diet and the other receiving a combination of the formulated diet and 10,000 striped beakfish newly-hatched larvae every day in each tank.
Could the authors clarify why this specific concentration was selected and why varying feeding levels were not tested?]
Response 6: [We appreciate the reviewer’s thoughtful comment. The daily amount of 10,000 striped beakfish newly-hatched larvae per tank was determined based on our previous experience with kawakawa larval rearing and direct observations made during the present experiment. When this quantity was provided at 9:00 AM each day, most of the larvae were consumed by late afternoon, with only a small number remaining, if any. This indicated that 10,000 larvae per day represented an optimal feeding level—neither excessive nor insufficient—for the larval stage during the experimental period. Therefore, we selected this amount as a practical and appropriate feeding level for this study.
In addition, in the end of Discussion section, 4.1.2, the authors added the future requirement of examination of the appropriate timing and quantity of hatched larvae to kawakawa larvae as follows (line 418-420).
Further research is required to determine the optimal timing and feeding amount of newly hatched fish larvae for kawakawa larvae and juveniles, in order to better understand and enhance their growth performance.]
Comments 7: [Line 70; In kawakawa, strong piscivorous behavior, which has been reported on around 12 dph.
The authors initiated the feeding experiment between day 9 and day 12 post-hatching, despite reports indicating the onset of strong piscivorous behavior in kawakawa around 12 dph. Could the authors clarify the rationale for selecting this specific timeframe and explain why a longer feeding period was not considered?]
Response 7: [The timing of feeding hatched larvae to E. affinis larvae is determined based on their degree of development, which is monitored continuously. This timing is strongly influenced by the quality of the eggs and can vary, occurring earlier or later depending on the batch of fertilized eggs. Moreover, the rearing protocols—such as feeding schedules and feeding amount—can vary between experimental conditions and those designed for large-scale production.
The authors additionally stated it in Section 2.2.1 (line 123-124); This rearing of kawakawa larvae was carried out at Kindai University as part of a commercially oriented project.]
Comments 8: [Could the authors clarify the reason for selecting striped beakfish (Oplegnathus fasciatus) as live prey for kawakawa? Is it feasible to produce this species at a scale sufficient to support kawakawa aquaculture, and does its use present a reasonable economic cost for commercial application? This information should be included in the introduction.]
Response 8: [Explanations were added in the manuscript as in the response of authors’ to the previous reviewer’s comment.]
Comments 9: [The authors stated that digital images were taken and used to measure body length (Lines 167, 211, 221); however, no images are provided in the manuscript. I recommend including these images as supplementary materials to enhance clarity and transparency for readers.]
Response 9: [We included it as Fig. 4.]
Comments 10: [Please reduce the redundant information in Sections 2.2 and 2.3.]
Response 9: [The authors revised 2.2 and 2.3 according to the reviewer’s comment as follows.
2.2. Experiment 1: Effects of feeding newly-hatched larvae on the growth and survival of kawakawa larvae
2.2.1. Test fish and rearing methods
The test fish used in this experiment were obtained from fertilized eggs produced by hormone-induced spawning of kawakawa broodstock (both males and females, hatched in 2020 and 2021) reared at the Wakayama Prefectural Fisheries Research Center. The fertilized eggs were stocked at a density of 7,540 eggs per tank in ten indoor 1 m³ circular tanks and reared until 12 days post-hatch (dph). This rearing of kawakawa larvae was carried out at Kindai University as part of a commercially oriented project .The normal hatching rate of the fertilized eggs was 83.2%.
During the rearing period, water quality was measured daily at 08:00 and 13:00. During the experiment, the dissolved oxygen (DO) level ranged from 83.6% to 107.6%. The pH values were between 8.1 and 8.4, while salinity levels (EC300, YSI, Maryland) ranged from 31.3 to 32.4 ppt, all within appropriate ranges. Furthermore, throughout the experiment, the smart larvae exhibited no signs of disease.
The rearing water temperature was maintained at 26.0°C throughout the experiment. An illuminance of 2,000 lx was provided by fluorescent, with a photoperiod of 13L:11D (light from 07:00 to 20:00).The aeration rate was set at 150 ml/min each during the daytime and two air stones (350 ml/min each) were added at night across all experimental tanks.
Kawakawa larvae were fed rotifers from 3 dph until the end of the experiment. The initial feeding density was 10 individuals/ml at 3 dph, and the feeding amount was increased in accordance with larval growth. The feeding rate was increased from 10 to 12 individuals/mL as the larvae grew. The rotifers were enriched with EPA- and DHA-fortified freshwater Chlorella (Hyper Gloss (Marinetech Co. Ltd., Aichi; Super Fresh Chlorella V12, Chlorella Industry Co., Tokyo), taurine (Taurine CJ, Japan Nutrition Co., Tokyo), and DHA-algae-concentrated enrichment agents for rotifers (Marin Gross EX, Marine Tech Co., Aichi). From 9 days post-hatching (dph), in addition to rotifers, each tank was supplied daily with 10,000 newly-hatched larvae of striped beakfish (Oplegnathus fasciatus), as well as a formulated diet (Magokoro, Nisshin Marubeni Feed Co., Tokyo). The seawater used for rearing was UV-sterilized and filtered, with the addition of with EPA- and DHA-concentrated Chlorella (Super Fresh Chlorella spV-12, Chlorella Industry Co., Tokyo) to maintain a density of 50 × 10⁴ cells/ml, adjusted according to the water supply volume.
Additionally, a water quality conditioner (Fish Green, Green Culture Co., Toyama) was applied at a rate of 15 g per tank before stocking eggs, and another 15 g per tank was added daily after feeding. Surface skimming to remove oil films was conducted at 08:30, 12:00, 15:00, and 16:30 across all experimental tanks.
2.2.2. Experimental groups
From 1 dph to 8 dph, all tanks were maintained under identical conditions. Starting from 9 dph, two experimental groups were established based on the presence or absence of newly-hatched larvae: formulated diet Group (F Group) – fed only a formulated diet, and formulated diet + newly-hatched larvae Group (FL Group) – fed a combination of a formulated diet and striped beakfish (Oplegnathus fasciatus) newly-hatched larvae. Each group consisted of five replicates.
2.2.3. Sampling and proportional measurement of larvae and juveniles
. Each day, 30 larvae or juveniles were randomly sampled from each tank, nd measured body length (BL; notochord length (NL) or standard length (SL) under a microscope.
2.2.4. Method for estimating survival rate
On the final day of the rearing trial, all larvae remaining in the tank were collected and the total number of larvae was counted.
2.2.5. Statistical Analysis
Statistical analyses were conducted using the statistical software EZR [48]. Differences between experimental groups were tested using analysis of variance (ANOVA), with a significance level set at 5% (p < 0.05). For body length (BL), a t-test was performed, while variance in length was assessed using an F-test. Survival rates were analyzed using the Mann-Whitney U test.
2.3. Experiment 2: relative growth from larval to juvenile stages in kawakawa
2.3.1. Test larvae and juveniles and their rearing methods
The test fish used in this experiment were obtained from fertilized eggs as the same source of Experiment 1. 100,000 fertilized eggs were stocked in each of two indoor 30 m³ circular tanks for rearing.
During the rearing period, dissolved oxygen (DO) levels, ranged from 100% to 120%, remaining within the appropriate range. Water temperature was maintained at 26.5°C until 4 dph, after which it was adjusted to 27.0°C from 5 dph onward and maintained until the end of the experiment. An illuminance of 2,000–5,500 lx was provided, with a photoperiod of 12L:12D (light from 06:00 to 18:00).
From 2 to 12 dph, kawakawa larvae were fed rotifers. The rotifers were enriched with EPA- and DHA-fortified freshwater Chlorella. From 9 to 22 dph, in addition to rotifers, larvae were provided with striped beakfish (Oplegnathus fasciatus) newly-hatched larvae sourced from the Kindai University Aquaculture Technology and Production Center Oshima Branch, with a feeding amount of 25 billion newly-hatched larvae per tank. From 7 dph until offshore transfer, a formulated diet (Magokoro, Nisshin Marubeni Feed Co., Tokyo) was provided in appropriate amounts.
The seawater used for rearing was UV-sterilized and filtered, with an appropriate amount of concentrated Chlorella algae (Fresh Chlorella V-12, Chlorella Industry Co., Tokyo) added as needed.
2.3.2. Sampling and measurement of larvae and juveniles
Each day, 30 larvae or juveniles were randomly sampled from each tank, anesthetize with eugenol and then euthanized in ice water, and digital images were taken, and the lengths of body parts were measured from the images. For absolute growth assessment, notochord length (NL) and standard length (SL) were measured and these were termed ‘body length (BL)’. For relative growth analysis, the following morphological parameters were evaluated: total length (TL), body length (BL), preanal length (PAL), head length (HL), snout length (SnL), body height (BH), head height (HH), caudal peduncle depth (CPD), upper jaw length (UJL), and eye diameter (ED). The test fish were larvae and juveniles reared for industrial-scale fingerling production at the Kindai University Aquaculture Technology and Production Center. A total of 10 to 30 individuals were randomly sampled at 1, 2, 3, 4, 5, 6, 10, 11, 13, 15, 17, 18, and 21 dph. The developmental stages of kawakawa were also referenced according to Kendall (1984) [49].
2.3.3. Statistical Analysis
Statistical analysis was performed using EZR [48]. Analysis of covariance (ANCOVA) was conducted to compare the differences in the slopes of the regression lines, with a significance level set at 5% (p < 0.05) in the analysis of kawakawa relative growth.]
Comments 11: [The authors seem to indicate that different amounts of striped beakfish were used in Sections 2.2 and 2.3. It is unclear why the same feeding protocol and management were not applied in both experiments.]
Response 11: [The rearing of kawakawa larvae and juveniles in Experiment 2 was conducted at Kinki University following a protocol designed for large-scale commercial production. Since commercial-scale production involves constraints not present in experimental rearing, the protocol differs from that used in controlled laboratory settings. The authors stated that the rearing conducted in Experiment 2 was carried out with the aim of commercial mass production in the section 2.2.1, Line 125 “This rearing of kawakawa larvae was carried out at Kindai University as part of a commercially oriented project”.]
Comments 12: [In Section 3.1.2 (Survival Rate), the authors present both the average survival rate across all replicate tanks and data from individual tanks, shown in two separate graphs. However, the individual tank data is not clearly described. The reason for presenting the data separately should be clarified. In addition, the figure legend does not adequately describe the two graphs. A clearer explanation is needed to help readers interpret the data.]
Response 12: [The reason we presented both the average survival rate and the individual tank data in separate graphs was to show that there were no tanks within the same treatment group that exhibited an exceptionally low survival rate compared to the others. By displaying the data in this way, we aimed to demonstrate the consistency of survival across replicate tanks. This also supports our position—raised in response to another reviewer comment—that the observed results were not due to external factors such as rearing environment issues or disease-related mass mortality, but were instead attributable to the experimental conditions, particularly the diet.
Moreover, presenting the individual tank data contributes to the transparency of our experimental results and allows readers to more fully evaluate the variability within treatment groups.
We have revised the figure legend to more clearly explain the content and purpose of the two graphs to help readers better interpret the data, and the contents of the figure was referred to in the 3.2.2 “None of the tanks exhibited an abnormally low survival rate that would suggest mass mortality of smart larvae caused by disease or adverse environmental factors.”
In addition
Figure 2. Survival rates in the formulated diet and in the formulated diet and live fish. Larvae group in Experiment 1.
was revised as
Figure 2. Survival rates in the formulated diet group (F group) and the formulated diet plus live fish group (FL group). The upper panel shows the average survival rate for each experimental group, while the lower panel presents the survival rates in individual rearing tanks (five replicates per group). ]
Comments 13: [The titles of Sections 3.1 and 3.2 do not clearly reflect the specific focus of this experiment. Revising the section headings to better align with the content would improve clarity.]
Response 13: [The authors revised the titles according to the reviewer’s comment as follows,
3.1 Experiment 1→3.1 Experiment 1: effects of feeding newly-hatched larvae on the growth and survival
3.2 Experiment 2→3.2 Experiment 2: relative growth]
Comments 14: [Please revise the order of presentation so that tables and figures appear in the sequence in which they are cited in the text—for example, Table 1 is placed before Figure 3, but Figure 3 is mentioned earlier in the manuscript.]
Response 14: [We sincerely thank the reviewer for pointing out the inconsistency between the order of figure and table citations in the text and their placement in the manuscript. In accordance with this valuable suggestion, we have revised the manuscript so that all tables and figures now appear in the order in which they are first mentioned in the main text. Specifically, Table 1 has been repositioned to follow Figure 3, as Figure 3 is cited earlier in the manuscript. We appreciate your attention to detail, which has helped improve the clarity and readability of our manuscript.]
Comments 15: [The authors should separate Figure 3 into Figures 3a and 3b and revise the corresponding references in the manuscript text accordingly.]
Response 15: [The authors separated Figure 3 into Figures 3a and 3b and revised the figure caption and the corresponding references in the manuscript text accordingly.
Figure 3. Absolute growth of body length (notochord length or standard length) of. kawakawa larvae and juveniles, their feeding regime, and developmental stage in Experiment 2. Vertical lines indicate ±SD (n=15-30).
was revised as
Figure 3. Absolute growth of body length (notochord length or standard length) of. kawakawa larvae and juveniles, their feeding regime, (upper panel; Figure 3a) and developmental stage (lower panel; Figure 3b) in Experiment 2. Vertical lines indicate ±SD (n=15-30).]
Comments 16: [Section 3.2.3 lacks a description of the results presented in Table 2. Please include relevant text to interpret and explain these findings.]
Response 16: [Since the content of Table 2—excluding the data on kawakawa larvae—consists of previously reported characteristics of other scombrid species, the authors deemed it more appropriate to address these points in the Discussion section rather than in the Results. Accordingly, they have discussed them in the Discussion.]
Comments 17: [Please avoid citing figure and table numbers in the Discussion. Focus on interpretation rather than result description.]
Response 17: [We appreciate the reviewer’s suggestion to avoid citing figure and table numbers in the Discussion section and to focus on interpretation. Nevertheless, we consider it important to reference specific tables and figures where appropriate, in order to clearly indicate the basis for our interpretations. We believe this approach helps ensure clarity and transparency in the discussion of our findings.]
Comments 18: [Lines 313–322 repeat results. Please revise to focus on interpretation and review the entire Discussion.]
Response 18: [It was revised.]
Comments 19: [Why do Lines 232–247 focus on increased DHA and growth when the study aims to evaluate the effect of striped beakfish supplementation on kawakawa growth? Please clarify this discrepancy. The manuscript title highlights feeding newly-hatched larvae, so the content should align with this focus. If discussing DHA, the authors should provide details on DHA content in the diet or proximate composition of the fish after feeding Kawakawa.]
Response 19: [In Discussion Section 4.1.1, the authors examined the relationship between DHA content in striped beakfish larvae and the growth acceleration observed in kawakawa larvae. This was discussed in the context of similar relationships reported in Pacific and Atlantic bluefin tuna. These comparisons provide a solid basis for discussing the potential role of striped beakfish hatched larvae supplementation in promoting the accelerated growth of kawakawa larvae fed with these larvae.]
Comments 20: [Line 353-355 “This suggests that size variation among individuals increased when hatched larvae were included in the diet, as some larvae grew faster while others did not.”
How did the authors ensure this size difference was caused by the diet rather than inconsistencies or poor feeding technique? Please clarify.]
Response 20: [Thank you for your valuable comment. In this rearing experiment, the only difference between the experimental and control groups was the feeding of hatched larvae. All other conditions, including the feeding method and tank management, were standardized across treatments. Therefore, we consider it reasonable to interpret the size variation observed in the smart larvae as a result of the inclusion of hatched larvae in the diet. In addition, although the same batch of eggs was used for the rearing of experimental group and control group, the growth was different between the groups.]
Comments 21: [The sentence in Lines [355-358] is grammatically unclear and needs revision for clarity. For example, the phrase “which made it appear as if growth had accelerated. hatched larvae” is incomplete and confusing. Please rewrite for better flow and accuracy.]
Response 21: [The authors revised this sentence by deleting” hatched larvae”.]
Comments 22: [Section 4.2, “Relative growth of kawakawa larvae and juveniles,” is unclear. Please clarify the main points and improve the explanation for better understanding.]
Response 22: [The authors added the summary of discussion “This section aims to clarify the characteristics of relative growth in kawakawa larvae and juveniles, in relation to their absolute growth. It highlights that larger scombrid species, such as tunas, tend to prioritize growth over development compared to smaller scombrids like kawakawa. Furthermore, even among scombrids, relative growth patterns vary, and based on the relationship between mouthpart development and piscivory—a key feature of scombrid fishes—these species can be categorized into three distinct groups.” following the first sentence “ The following discussion on inter-species comparisons of growth patterns in scombrid species references studies on Pacific bluefin tuna (Thunnus orientalis; [15]), yellowfin tuna (Thunnus albacares; [46]), striped bonito (Sarda orientalis; [15]), and chub mackerel (Scomber japonicus; [47]).” (line 423-432)]
Comments 23: [The Discussion should focus on interpreting the results from this study, specifically comparing the effects of feeding live feed between the control and treatment groups. However, it currently emphasizes comparisons with other scombrid larvae. Please revise to prioritize your own study’s findings.]
Response 23: [The authors consider that, in order to clarify the relative growth characteristics of kawakawa larvae and juveniles, comparison with other scombrid species represents a reasonable and effective approach. Combining the findings of this study with comparative data from related species allows for a deeper understanding of the developmental traits unique to kawakawa.]
Comments 24: [All scientific names should be italicized throughout the manuscript. Please check for consistency.]
Response 24: [We italicized all the scientific name in the manuscript.]
Reviewer 2 Report
Comments and Suggestions for Authors
Dear Authors, the present manuscript requires minor modifications prior to publication. Please find below my suggestions and comments.
Title: « Effects of Feeding Newly-Hatched Larvae on the Growth, Survival, and Relative Growth Patterns of Kawakawa (Euthynnus affinis) Larvae and Juveniles ».
Would it be conceivable to gather “Growth” and “Relative Growth patterns” together as “Growth Patterns”? The title would be less redundant.
L.23: “Fertilized eggs were reared…” I think it would be valuable information here to specify which fish species you are talking about. In the first part of the abstract you mention two fish species, therefore when you start the sentence with “Fertilized eggs were reared…” I assume that you are talking about kawakawa but this is uncertain.
L.27-28: Could you indicate how long took the feeding experiment.
L.51: “…including those in 9 [9] and 10 [10] in Japan.” Could you explain here what is 9 and 10?
L.80: Could you explain what is 10s?
L.86: “…relative growth patterns…” Could you describe how is defined a relative growth pattern?
L.118: “…hormone-induced spawning…” Could you specify the manufacturer and the dose per kg of fish? Same with L.187.
L.121: “…reared until 12 days post-hatch (dph).” Was this rearing also performed at 26°C? If yes, please specify it earlier than L.130.
L.166: “…anesthetized with eugenol…” Please indicate concentration of eugenol and manufacturer. Same with L.211.
L.168:” For image acquisition…” Please indicate the camera used. Same with L.211, L.221.
L.220: “…at 1, 2, 3, 4, 5, 6, 10, 11, 13, 15, 17, 18, and 21 dph, euthanized using ice…” Were the fish anesthetized with eugenol prior to euthanasia?
L.246-Figure1: Could you make the symbols for the Formulated feed and Hatched larvae+Formulated feed bigger? They should be at least twice bigger than the thickness of the curve to be easily identified. In addition, a “t” is missing on the legend of Hatched larvae+Formulated feed.
L.251: “The survival rate on 13 dph was…” Could you replace “on” with “at” to be more consistent with the whole manuscript?
L.254-Figure 2: Please label the y axis of the two charts. In addition, the two charts should be labelled A and B.
L.260: “Growth accelerated around 13 dph.” Could you specify the figure you are referring to here? I assume we can witness the growth acceleration in Figure 3, but it must be made clear.
L.279-Figure 3: Please label the two figures as Figure 3A and Figure 3B.
L.294-Figure 4: Please label the figures from Figure 4A to Figure 4I. Please check the font size uniformity. It seems that “Head height” graph title is smaller than the others graph titles.
L.350: “…due to the species’ strong piscivorous tendencies.” Could you replace species’ with specie’s?
L.351: “…survival rate in F Group…” Could you update the graph to indicate F group in the x axis? This would make the manuscript clearer. Same comment for FL.
L.516-522: “…in Pacific bluefin tuna (Thunnus orientalis),…” “…Pacific bluefin tuna (Thunnus orientalis).” At the first mention of a species in the manuscript, the common name and the scientific name in italic should be indicated (e.g. Pacific bluefin tuna (Thunnus orientalis)). However, the for the subsequent mention of the fish species, the common name only would be sufficient. This comment applies to all fish species in the whole manuscript.
Author Response
Comments 1: [Title: « Effects of Feeding Newly-Hatched Larvae on the Growth, Survival, and Relative Growth Patterns of Kawakawa (Euthynnus affinis) Larvae and Juveniles ». Would it be conceivable to gather “Growth” and “Relative Growth patterns” together as “Growth Patterns”? The title would be less redundant.]
Response 1: [We agree with the reviewer’s suggestion and have revised the title accordingly to reduce redundancy by combining “Growth” and “Relative Growth Patterns” into “Growth Patterns.”
« Effects of Feeding Newly-Hatched Larvae on the Growth, Survival, and Growth Patterns of Kawakawa (Euthynnus affinis) Larvae and Juveniles ».]
Comments 2: [L.23: “Fertilized eggs were reared…” I think it would be valuable information here to specify which fish species you are talking about. In the first part of the abstract you mention two fish species, therefore when you start the sentence with “Fertilized eggs were reared…” I assume that you are talking about kawakawa but this is uncertain]
Response 2: [“Fertilized eggs were reared ---“
was revised to
“Fertilized eggs of kawakawa were reared ------”]
Comments 3: [L.27-28: Could you indicate how long took the feeding experiment.]
Response 3: [In Line 27, the experimental period was described.]
Comments 4: [L.51: “…including those in 9 [9] and 10 [10] in Japan.” Could you explain here what is 9 and 10?]
Response 4: [Thank you for pointing this out. The mention of "9 [9] and 10 [10]" was the result of a formatting error. It has been corrected in the revised manuscript.]
Comments 5: [L.80: Could you explain what is 10s?]
Response 5: [Thank you for pointing this out. The mention of "9 [9] and 10 [10]" was the result of a formatting error. It has been corrected in the revised manuscript.]
Comments 6: [L.86: “…relative growth patterns…” Could you describe how is defined a relative growth pattern?]
Response 6: [We appreciate the reviewer’s comment. To improve clarity, we have revised the relevant sentence to better reflect this definition as follows.
“Understanding such relative growth patterns of larvae and juveniles is crucial for optimizing rearing techniques -------”
→Understanding such relative growth patterns—defined as changes in the growth rates of specific body parts in relation to overall body length, including transitions among positive allometry, isometry, and negative allometry—is crucial for optimizing rearing techniques by providing insights into functional transformations across developmental stages. (line 89-93)]
Comments 7: [L.118: “…hormone-induced spawning…” Could you specify the manufacturer and the dose per kg of fish? Same with L.187.]
Response 7: [2.2.1. Test fish and rearing methods 119
Line 118 “The test fish used in this experiment were obtained from fertilized eggs produced by 120 hormone-induced spawning of kawakawa broodstock----“
was revised to provide necessary information as
The test fish used in this experiment were obtained from fertilized eggs produced by 120 hormone-induced spawning of kawakawa broodstock using HCG (500 IU/kgBW, GONATROPINⓇ3000, ASKA Animal Health Co., Ltd., Tokyo) (line 121-124)]
Comments 8: [L.121: “…reared until 12 days post-hatch (dph).” Was this rearing also performed at 26°C? If yes, please specify it earlier than L.130.]
Response 8: [The sentences
During the rearing period, water quality was measured daily at 08:00 and 13:00. The dis-128 solved oxygen (DO) levels measured using a DO meter (550A, YSI, Maryland), ranged 129 from 83.6% to 107.6%. The pH values (D-24, HORIBA, Kyoto) were between 8.1 and 8.4, 130 while salinity levels (EC300, YSI, Maryland) ranged from 31.3 to 32.4 ppt, all within ap-131 propriate ranges.
were revised as
Larvae were reared at 26°C, and during the rearing period, water quality was measured daily at 08:00 and 13:00. The dis-128 solved oxygen (DO) levels measured using a DO meter (550A, YSI, Maryland), ranged 129 from 83.6% to 107.6%. The pH values (D-24, HORIBA, Kyoto) were between 8.1 and 8.4, 130 while salinity levels (EC300, YSI, Maryland) ranged from 31.3 to 32.4 ppt, all within ap-131 appropriate ranges. (line 126-131)]
Comments 9: [L.166: “…anesthetized with eugenol…” Please indicate concentration of eugenol and manufacturer. Same with L.211.]
Response 9: [It was revised as “eugenol (FA100,Bussan Animal Health Co., Ltd.)”(line 196)]
Comments 10: [L.168:” For image acquisition…” Please indicate the camera used. Same with L.211, L.221.]
Response 10: [It is revised.]
Comments 11: [L.220: “…at 1, 2, 3, 4, 5, 6, 10, 11, 13, 15, 17, 18, and 21 dph, euthanized using ice…” Were the fish anesthetized with eugenol prior to euthanasia?]
Response 11: [Yes, and it was stated in Line 198.]
Comments 12:[ L.246-Figure1: Could you make the symbols for the Formulated feed and Hatched larvae+Formulated feed bigger? They should be at least twice bigger than the thickness of the curve to be easily identified. In addition, a “t” is missing on the legend of Hatched larvae+Formulated feed.]
Response 12: [The symbols were made bigger in Figure 1.]
Comments 13: [L.251: “The survival rate on 13 dph was…” Could you replace “on” with “at” to be more consistent with the whole manuscript?]
Response 13: [Revised according to the comment.]
Comments 14: [L.254-Figure 2: Please label the y axis of the two charts. In addition, the two charts should be labelled A and B.]
Response 14: [In the y axis, (%) was changed to Survival rate (%), and two charts were labelled A and B with the revision of the figure legend.]
Comments 15: [L.260: “Growth accelerated around 13 dph.” Could you specify the figure you are referring to here? I assume we can witness the growth acceleration in Figure 3, but it must be made clear.]
Response 15: [Referred figure was specified as follows.
Growth accelerated around 13 dph.→Growth accelerated around 13 dph (Fig.3).]
Comments 16: [L.279-Figure 3: Please label the two figures as Figure 3A and Figure 3B.]
Response 16: [Two figures in Fig. 3 were labeled as Figure 3a and Figure 3b, and referred in the text as they are.]
Comments 17: [L.294-Figure 4: Please label the figures from Figure 4A to Figure 4I. Please check the font size uniformity. It seems that “Head height” graph title is smaller than the others graph titles.]
Response 17: [The authors do not agree with the labeling the figures in Fig. 4, because the authors think the readers can find the figure easily to which the authors mention in Results and Discussion section. The inconsistency of the font size was revised.]
Comments 18: [L.350: “…due to the species’ strong piscivorous tendencies.” Could you replace species’ with specie’s?]
Response 18: [The word was replaced.]
Comments 19: [L.351: “…survival rate in F Group…” Could you update the graph to indicate F group in the x axis? This would make the manuscript clearer. Same comment for FL.]
Response 19: [Fig 2. was revised.]
Comments 20: [L.516-522: “…in Pacific bluefin tuna (Thunnus orientalis),…” “…Pacific bluefin tuna (Thunnus orientalis).” At the first mention of a species in the manuscript, the common name and the scientific name in italic should be indicated (e.g. Pacific bluefin tuna (Thunnus orientalis)). However, the for the subsequent mention of the fish species, the common name only would be sufficient. This comment applies to all fish species in the whole manuscript.]
Response 20: [Revised according to the reviewer’s comment.]

Reviewer 3 Report
Comments and Suggestions for Authors
The authors of this article address key issues related to nutrition and feeding in aquaculture, highlighting current challenges such as the lack of specialized feeds tailored to specific fish species. Such feeds could fulfill all the nutritional requirements of the fish, closely replicate their natural diet, and simultaneously have a positive impact on health, welfare, and production performance. Furthermore, the authors present an analysis of the relative growth of Kawakawa larvae and juveniles in comparison to other tuna species, offering valuable insights for the further development and cultivation of this species. However, comments and proposed changes that could make the text clearer have been included below.
- ,, Full-cycle aquaculture systems have been developed at 50 fisheries research institutions, including those in 9 [9] and 10 [10] in Japan.’’ Please check the numbering of the items in the bibliography - in the text there was most likely a repetition of the citation, both items 9 and 10 are in brackets and without brackets.
- The authors describe the feeding system in quite detail, but there is no information on the frequency of feeding, i.e. how many times a day the fish were fed.
- The authors obtained results in the first part of the experiment refer to body length, it would be necessary to consider body mass and relate them, for example, to production results such as feed conversion. This could comprehensively illustrate the differences in groups.
- In the second part of the experiment in resalts there is no indication whether there were statistically significant differences, and if so, in which parameters.
- Consideration should be given to presenting other authors' results on the feeding of Artemia nauplii in a table in the discussion section, which will increase the accessibility and clarity of the text.
- In line 357 there is a period missing at the end of the sentence.
- From lines 358 to 385 there is a description of the development of the stomach, the activity of digestive enzymes. How does this relate to the results presented by the authors? Consideration should be given to including results concerning the development of the digestive system, e.g. histology and activity of digestive enzymes.
- Part of the discussion 4.2. "Relative growth of kawakawa larvae and juveniles" should be presented in a table, where the results obtained by the authors can be compared with those of other authors. Such a comparison would be more readable.
- Should pay attention to the font style. Some of the headings are italic, some are not. You need to systematize it according to the journal's guidelines.
- Please note the italics in the names of fish species, as they are not used throughout the text (especially in the discussion section).
Author Response
Comments 1: [Full-cycle aquaculture systems have been developed at 50 fisheries research institutions, including those in 9 [9] and 10 [10] in Japan.’’ Please check the numbering of the items in the bibliography - in the text there was most likely a repetition of the citation, both items 9 and 10 are in brackets and without brackets.]
Response 1: [Revised.]
Comments 2: [The authors describe the feeding system in quite detail, but there is no information on the frequency of feeding, i.e. how many times a day the fish were fed.]
Response 2: [The sentences of Line 141-143
Kawakawa larvae were fed rotifers from 3 dph until the end of the experiment. The initial feeding density was 10 individuals/ml at 3 dph, and the feeding amount was increased in accordance with larval growth.
were revised to provide informations on the frequency of feeding as
Kawakawa larvae were fed rotifers from 3 dph until the end of the experiment. The initial feeding density was 10 individuals/ml at 3 dph, where rotifers were supplied at 08:00AM every morning and added to maintain the target density, and the feeding amount was increased in accordance with larval growth (line 136-139).
The sentence of Line 148-152
each tank was supplied daily with 10,000 newly-hatched larvae of striped beakfish (Oplegnathus fasciatus),
was revised as
each tank was supplied with 10,000 newly-hatched larvae of striped beakfish (Oplgnathus fasciatus) once every morning, obtained from the Kindai University Aquaculture Technology and Production Center Oshima Branch, as well as a formulated diet fed at any time (Magokoro, Nisshin Marubeni Feed Co., Tokyo). (line 143-147)]
Comments 3: [The authors obtained results in the first part of the experiment refer to body length, it would be necessary to consider body mass and relate them, for example, to production results such as feed conversion. This could comprehensively illustrate the differences in groups.]
Response 3: [We appreciate the reviewer’s insightful comment. Unfortunately, body mass data were not collected during the present study. We acknowledge that incorporating such measurements, along with indicators like feed conversion ratio, would provide a more comprehensive understanding of group differences. We consider this an important aspect for future research and will address it in our subsequent studies.]
Comments 4: [In the second part of the experiment in resalts there is no indication whether there were statistically significant differences, and if so, in which parameters.]
Response 4: [We thank the reviewer for this valuable comment. In the section 2.3.3., the authors stated the use of statistical method in the relative growth analysis of kawakawa larvae and juveniles. We revised the section 2.3.3. as follows to make readers understand this clearly.
2.3.3. Statistical Analysis
Statistical analysis was performed using EZR [48]. To determine whether the slopes of all regression lines in the relative growth analysis of kawakawa larvae and juveniles significantly differed before and after the inflection points, analysis of covariance (ANCOVA) was conducted. A significance level was set at 5% (p < 0.05), and statistically significant differences were found in all comparisons of slopes before and after the inflection points. (line 210-216)]
Comments 5: [Consideration should be given to presenting other authors' results on the feeding of Artemia nauplii in a table in the discussion section, which will increase the accessibility and clarity of the text.]
Response 5: [We appreciate the reviewer’s suggestion. In Section 4.1.1 of the Discussion, we have cited previous studies highlighting the nutritional limitations of Artemia nauplii, particularly their low DHA content. We also pointed out that newly hatched kawakawa larvae contained approximately 23 times more DHA than Artemia nauplii, which likely contributed to the enhanced growth observed in the larvae. Therefore, we believe the current presentation sufficiently addresses the comparative aspect, although in narrative rather than tabular form.]
Comments 6: [In line 357 there is a period missing at the end of the sentence.]
Response 6: [A period was added.]
Comments 7: [From lines 358 to 385 there is a description of the development of the stomach, the activity of digestive enzymes. How does this relate to the results presented by the authors? Consideration should be given to including results concerning the development of the digestive system, e.g. histology and activity of digestive enzymes.]
Response 7: [Lines 358 to 385 were revised to clarify the relationship between kawakawa larval piscivory and the development of stomach and gastric glands.
In the present study, we observed that kawakawa larvae began feeding on newly hatched striped beakfish at 9 dph, when they were approximately 6 mm TL and had reached the flexion stage. This timing appears to coincide with the initial development of gastric function, as reported by Duy Khoa et al. (2021) [69], who noted the onset of pepsin activity around 10 dph. This suggests that the development of the stomach and gastric glands is a key prerequisite for the initiation of piscivory in kawakawa larvae. The ability to digest complex prey such as fish larvae likely contributed to the rapid growth observed in the treatment group.
Previous studies in related species, including Thunnus orientalis, T. thynnus, T. albacares, and Scomberomorus niphonius, also support the notion that the onset of pepsin activity and gastric gland formation marks a physiological shift toward piscivory [13–14, 46, 70–71]. Our findings, while not directly assessing histology or enzyme activity, align with these developmental timelines and suggest that kawakawa may exhibit relatively early gastric development compared to other scombrids, which may underlie their early transition to piscivory and associated growth acceleration. (line 380-387)]
Comments 8: [Part of the discussion 4.2. "Relative growth of kawakawa larvae and juveniles" should be presented in a table, where the results obtained by the authors can be compared with those of other authors. Such a comparison would be more readable.]
Response 8: [The authors think Table 1, 2 and Figure 4 are sufficiently present the relative growth of kawakawa larvae and juveniles. Actually the authors could collected enough materials for the comparison of the relative growth in scombridae from the published papers which presented the same kind of materials as our paper.]
Comments 9: [Should pay attention to the font style. Some of the headings are italic, some are not. You need to systematize it according to the journal's guidelines.]
Response 9: [Revised according to the comment.]
Comments 10 [Please note the italics in the names of fish species, as they are not used throughout the text (especially in the discussion section).]
Response 10: [Revised according to the comment.]

Round 2
Reviewer 1 Report
Comments and Suggestions for Authors
The authors have addressed all my comments thoroughly and clearly. The revised manuscript is significantly improved.